# Analyses of the Effects of Electrolyte and Electrode Thickness on High Temperature Proton Exchange Membrane Fuel Cell (H-TPEMFC) Quality

**DOI:** 10.3390/membranes13010012

**Published:** 2022-12-22

**Authors:** Shubham Manoj Nawale, Mangaliso Menzi Dlamini, Fang-Bor Weng

**Affiliations:** Department of Mechanical Engineering, Fuel Cell Centre, Yuan Ze University, Chung-Li District, Taoyuan City 32003, Taiwan

**Keywords:** high-temperature proton exchange fuel cell, Impedance, electrode, electrolyte, thickness

## Abstract

Researchers have been striving to minimize proton exchange membrane fuel cell components thickness. This is believed to reduce the losses (active losses, ohmic losses and mass transfer losses) associated with this cell. In this study, we numerically analyze the electrodes and electrolyte thickness effects on high-temperature proton exchange membrane fuel cell (H-TPEMFC) performance. COMSOL Multiphysics is adopted to model both the impedance spectroscopy and polarization of the cell. Increased cell catalyst layer (thick electrode) improves the overall cell performance by ±10%, because of the improved reaction rate. It presents 0.89 mol m^−3^ lesser oxygen compared to that of the thin electrode cell. On the contrary, thick cell electrodes come with increased mass transport loss. The high reaction rate is also confirmed by the high amount of generated water, which is 0.42 mol m^−3^ higher than that of thin electrode cell. The experiment used to set the modeling parameter renders results with only less than 5% discrepancy to the modeling results. Also revealed is that over a limited range, electrolytes thickness variation has negligible effects on H-TPEMFC performance.

## 1. Introduction

The development of fuel cells is governed by cost, material properties, efficiency, durability, environmental friendliness and system simplicity. Fuel cells are promising clean energy supply devices. They come in different forms with different operating conditions and principles. The presented fuel cell is H-TPEMFC, which is basically a proton exchange membrane fuel cell (PEMFC) requiring high temperature operating conditions. This cell is characterized by high conversion efficiency, high operating temperature, clean emissions, and high power density. Comparing the H-TPEMFC with a lithium battery, fuel cell keeps running, so long as there are enough reactants supply. On another note, H-TPEMFC reactants are readily available compared to fossil fuels. This cell is less sensitive to fluid flow compared to low temperature proton exchange membrane fuel cell. This is because most of the fluid is in gas phase. To gain full market recognition, the Fuel Cell industry is currently battling with high weight, short lifespan, high cost, and low power density. Researchers are working on all the cell components to alleviate the aforementioned challenges. A fuel cell can either be operated as a single (unit) cell or stack, depending on the power demand. Stacked cells come with a number of challenges. As the stack size increases, mass transfer losses increase too. Some studies have stated that in a stack, margin cells receive less reactants [1,2]. This is believed to depend on the inlet design and positioning, although these claims required further investigation because it is not consistent with other studies [3,4]. Other studies show that this problem is encountered by the middle cells. Baik et al. [5,6] showed that if the reactants supply is well positioned, the cell will get the same reactants quantity, but they may still have some discrepancy in performance.

Studies have proven that the cell cathode can either be open or purge controlled [7,8,9]. Open cathodes have a number of advantages, e.g., there is no need for oxygen tanks, and their auxiliaries. Open cathodes also improve cell cooling [10,11]. On the other hand, it renders some drawbacks. Dust and cell dehydration is predominant. Open channels are only feasible on the cathode side as oxygen can be readily used from the ambient air, while it is impossible with the anode. This is because pure hydrogen is recommended for efficiency. Studies have even considered the implementation of dead-end channels [12,13], and hydrogen recycling [7,14] in the anode. Among other factors, the purging pressure settings consider the electrolyte mechanical strength and thickness. Increased electrolyte strength and thickness can accommodate a low purge rate, thus maximizing fuel utilization [15]. Weight and volume are being tackled through the introduction of metallic bipolar plates. Cost is being tackled mainly by the development of cheap catalyst materials. Durability is mainly addressed through the development of high strength electrolyte materials. PEMFCs are gaining acknowledgment in the automotive and UAV industries [16,17,18,19,20,21]. These industries demand lightweight cells with a high power convection rate. This study is focused on the development of a state-of-the-art H-TPEFC to be implemented in a UAV. This demands high-strength cells.

Resistance plays a prominent role in PEMFC. In addition to thickness, the ionic resistance in the electrolyte is dependent on temperature and humidity [15]. There are many components contributing to the overall cell impedance. Actually, almost every item in the cell system has a share in impedance. Jian et al. [22] implemented electrochemical impedance spectroscopy measurements to understand the conductivity and sustainability of a polypyrrole-graphene oxide/polypyrrole-camphorsulfonic acid bilayer composite coating (PPY-GO/PPY-CSA) on a 304 stainless steel bipolar plate. Park et al. [23] proposed a foam flow field membrane electrode assembly with a higher catalyst loading and thinner membrane thickness to achieve outstanding performance (current density > 2.0 A cm^−2^ at 0.6 V). In our study, we only model the effects of the electrode and membrane thickness. It is assumed that the cell electrodes are homogeneous; therefore, only one electrode is considered in the impedance spectroscopy. Considerable attribute to the impedance storms from the resistance of charge transfer and the oxide layer accumulating over the bipolar plate. Grandi et al. [24], analyzed catalyst influence on cell impedance. They concluded that the catalyst thickness contributes in the membrane conductivity through water retention.

All the aforementioned components considered in H-TPEMFC developments should never compromise efficiency. The main challenge usually encountered in addressing these factors is that most of the developments counter each other, e.g., bipolar plate coating compromises cost and conductivity, cost reduction compromises efficiency and quality. Researchers are putting more focus on metallic bipolar plate developments as it is believed that they have improved conductivity. Metallic bipolar plates can also withstand the notorious but lucrative automobile industry conditions and requirements. Corrosion is their main drawback.

To our knowledge, this is the first work that evaluates the most influential components in H-TPEMFC impedance between the electrode and electrolyte. This will consequentially simplify the PEMFC development to improving power density. Most researchers are targeting the influence of each of this components, independent. Others are only working on reducing the ICR, which is more influenced by bipolar plates morphology and materials. DOE’s has set an ultimate power density target of 850 W/L [24,25,26].

Researchers [27] confirmed a cell improvement in catalyst and membrane thickness reduction. They recommended a membrane thickness ranging from 20–60 μm. Other researchers [28] have concluded that even though the catalyst increment renders improved results, it is still not comparable to the Pt catalyst cost. They further discover that increasing the membrane thickness renders negative results. They investigated the catalyst, and membrane of up to 150 and 100 μm, respectively. Therefore, storms the need of a cheap catalyst materials. Zhang et al. [29] investigated the fluid flow in H-TPEMFC, and concluded that water vapor transport flux decreases with increasing membrane thickness. Increasing the pressure and humidity (60%) of the anode side causes any water vapor to crossover to the cathode side, which is beneficial to the cell performance. To avoid membrane damage, the cathode pressure should be at least 0.2 bar higher than that of the anode [29,30]. Xia et al. [31], investigated the effect of GDL thickness, and discovered that it is significantly effective in the cell performance. The optimum thickness range for both cathode and anode is 140–170 and 80–120 mm, respective. The porosity range is 35–45%. 

Xia et al. [32] investigated the catalyst thickness, and found that reaction rate increases with catalyst increment. They suggested that the conclusion that H-TPEMFC performance can be improved through reducing the catalyst thickness should be reviewed. That is among the motivations of this study. This study evaluates the more influential component in the cell, between the membrane and electrode thickness. Cost has been ignored. The novel contributions of this study includes: 1. the generated water analyses in H-TPEMFC; 2. comparing the electrode and membrane thickness influence; 3. evaluating the electrochemically consumed reactants in the cell; 4. deep understanding of material conductivity feasibility in covering up the losses associated with the membrane thickness. 

## 2. Materials and Methods

### 2.1. H-TPEMF Advantages

To set the parameters for the numerical analysis, an experiment was conducted with the conditions presented in Table 1 and Table 2, otherwise we adopted the parameters presented in our previous work [33,34]. The model and experimental conditions are the same, unless stated.

As aforementioned, H-TPEMFC suffers less water flooding because most of the fluid exist in gas phase. The cell components are made in such a way that they are less sensitive to hydration. On another note, less pressure is required to drive fluid in gas phase than in liquid phase. Unlike normal temperature PEMFC, H-TPEMFC cell requires a longer startup time. When conducting H-TPEMFC experiment, the cell needs to be cladded with an insulator, Figure 1a,b to minimize heat energy loss. The heating rate needs to be well controlled to alleviate degradation. 0.5 °C min^−1^ is the heating rate adopted in this work.

### 2.2. Modelling Set Up

In this study the geometry was prepared in Solidworks, and exported to COMSOL Multphysics for the simulation. Experiments have only been conducted to validate the numerical parameters. The serpentine channels shown in Figure 2 have been adopted for both the cathode and anode flow fields. The numerical analysis is mainly focused on the electrodes and electrolyte thickness influence on the overall H-TPEM fuel cell performance. Table 1 lists the cell geometry and conditions, and Table 2 lists the cell operating conditions adopted in both the experiment and modeling. The cells will be named according to the electrolyte and electrode thickness. Cell 3 is basically cell 1, with improved conductivity values. Its purpose is to see if it can compensate the reduced catalytic effects caused by the electrode reduction. The operation temperature for all the cells is set to 180 °C. The model mass flow inlet is governed by velocity, while the outlet is governed by pressure. Figure 3 shows the schematic diagram of the cells adopted in this study. Figure 3a portrays the thin electrode cell, while Figure 3b portrays the thick electrode cell. In both Figure 3a,b, the 1D figure portrays the half-cell (one electrode and the electrolyte) used for the impedance simulation. The electrode part combines both the GDL and the catalyst layer.

The simulation is conducted under the following assumption: 1. Homogeneous porosity in individual component; 2. Even gas distribution; 3. Fluid flow is laminar, with ignorable gravity; 4. No species cross-over; 5. Ideal gas mixture; 6. Slip condition is applied over the cell periphery with the exception of the inlet and outlet; 7. GDL is more porous compared to the catalyst layer; 8. Incompressible fluid; 9. Water exist in gas phase in the triple phase boundary. Ribeirinha et al. [35] and [24] Perng et al. [24] well present the source terms and the governing equations adopted in this work.

### 2.3. Components Geometry

*Bipolar plates*: Bipolar plates are responsible for the fluid flowing in and out of the cell. It also provides mechanical strength for the entire cell. Bipolar plates should have the following key qualities: corrosion resistance, chemical and mechanical stability, good current and thermal conduction. This part is the main component, accounting for weight and volume. That is the reason researchers are turning away from the conventional graphite bipolar plate, and focusing on metallic bipolar plates, although metallic bipolar plates suffer corrosion effects under the H-TPEMFC working environment. Bipolar plates come in different designs and shapes. There are: serpentine flow, straight channels, interdigitated flow, pin-type, bio inspired pattern, fine mesh. In addition to the general mass flow, bipolar plate design has been noted for its tremendous contribution to pressure drop, diffusion enhancements, and water drainage. The inlet and outlet also play a prominent role in active area utilization. In recent flow field developments, researchers have instituted foam in the channels [36,37]. Some have instituted partially blocked channels [38]. This study has adopted serpentine channels, machined on graphite plate. The major influence behind this selection is production cost and BOP simplicity. It is believed that some flow field demand the assistance of pressure booster to ensure reactants continuous flow. Serpentine flow field is more convenient for small area cells than large area cells.

*Gas diffusion layer (GDL)*: The GDL mainly ensures even reactant distribution and water removal in the triple phase boundary layer (TPB), current collection from the TPB, and provides mechanical strength for the cell MEA. The GDL has to be more porous than the catalyst layer. Studies have revealed an optimum overall cell porosity range of 30–40%. The GDL is usually made of carbon fiber, carbon paper or carbon cloth, although other materials are still under research. Although the GDL has a prominent thickness, it still has to be under reasonable value to avoid being waterlogged. When the GDL is too thick, it increases pressure drop. This component of the cell should be of good hydrophobicity to ensure that the produced water does not remain and cause flooding in the cell. To control the GDL hydrophobicity, carbon cloth or carbon paper is usually coated with fluorinated ethylene propylene (FEP) or polytetrafluoroethylene (PTFE) [39,40,41]. Extra attention needs to be paid when loading these materials, as they compromise the carbon conductivity. In addition, more hydrophobic materials can block the pores, thus affecting species diffusion, and generated water removal. The pore importance is further considered during the cell assembly load application. Lee et al. [42] adopted the radio frequency (RF) plasma treatment method to modify the GDL hydrophobic properties with CF_3_ CF_4_ and SF_6_. It appeared that CF_3_ presents superior results over the other components. RF is a good method to treat carbon, because it does not threaten the cell porosity and conductivity if well controlled.

*Electrode (catalyst and support)*: Electrodes play a prominent role in the reaction process. These are the solid bodies sandwiching the electrolyte, named the anode and the cathode. They contain the TPB. Each electrode constitutes the catalyst layer, which is traditionally platinum with a carbon layer (Teflon) that support the catalyst. Among reasons for carbon support adoption is the skyrocketing platinum prize. Carbon further presents acceptable conductivity. Platinum is added in small quantities, in nanoparticles. The nano-size maximizes its active surface area. Extra components have to be added to adhere the platinum particles to the carbon layer; otherwise they will be eroded away during fluid flow. Other materials have been tried, while others are still under research in an effort to replace expensive platinum. The electrode qualities are: 1. Current and thermal conductivity; 2. Chemical stability; 3. Thermal stability; 4. Porosity (although lower than that of the GDL). Electrodes should have a controlled thickness to minimize mass transport losses, while facilitating reaction. The electrode should have manageable hydrophobic properties, while it should remain hydrated enough to maintain good conductivity for itself and the electrolyte. Among the challenges faced with catalyst layers is agglomeration and delamination. At high pressure, catalyst particles get washed away. They get eroded from the carbon substrate, and either collect in the electrolyte surface, or get carried away through the drained water.

*Electrolyte thickness*: The electrolyte is a thin layer sandwiched by the electrodes in direct contact with the TPB. It is traditionally made of Nafion. This component should be dense enough to prevent fluid crossover. Its properties are: 1. High ionic/protonic conductivity, 2. High material density, 3. Thermal stability, 4. Chemical stability. Generally, Nafion perfectly operate at temperatures around 100 °C, although after modifications, it can withstand temperatures of up to 190 °C [43,44,45]. Numerous studies have been conducted to improve the electrolyte components. The catalyst thickness needs special attention. 

### 2.4. Temperature Effect

In relation to the components thickness, temperature increase helps in reduce the cell active and ohmic losses. The main challenge rendered by increased operating temperatures is material limitation, mainly the electrolyte. It also weakens catalyst adhesion to its support, making it more prone to agglomeration. It also promotes corrosion. Higher temperature increases conductivity, and improves catalytic activity. Extra high temperature may denature the catalyst. Temperature is among factors contributing to solid oxide cells (SOC) superiority [46]. Vogel-Fulcher-Tammann equation expresses the cell high activation energy demand at low temperatures, Equation (1) [47].
(1)$$\log\sigma =\log\sigma_{\infty }-\frac{E_{act}}{R\left(T-T_{0}\right)}$$

σ is the proton conductivity $\infty_{\sigma }$ is the pre-exponential factor, $E_{act}$ is the activation energy of the process underlying the DC-conductivity and R is the standard gas constant (8.314 J·mol^−1^·K^−1^). $T_{0}$ is the Vogel temperature. $\sigma_{\infty }$ is the pre-factor related to the limiting conductivity at high temperatures.

### 2.5. Protonic Conductivity

Teixeira et al. [48] used Equation (2) to portray proton conductivity ($\sigma_{p}$) of the electrolyte. Their results show improved conductivity and performance with temperature and relative humidity increment. It is the same reason why PEM cells should be kept hydrated.
(2)$$\sigma_{p}=\frac{L}{AR_{b}}$$

*L* is the distance through the electrolyte, *A* is the cross-sectional area, and $R_{b}$ is the bulk resistance.

### 2.6. Resistance and Losses

In Fuel cells, resistance is a serious issue. Efforts to reduce resistance are countered by cost effects, which are among serious issues that are forbidding fuel cell commercialization. For example, highly conductive materials and highly catalytic materials are all expensive. In fuel cells, there are three major losses [49,50,51]: kinetic losses, ohmic losses and mass transport losses, Figure 4. 

### 2.7. Model Description

Figure 5 shows the flowchart followed when conducting this study.

### 2.8. Grid Size

The maximum element size was set to 0.00365 m, minimum element size set to 4.57 × 10^−8^ m, with maximum element growth rate of 1.45, curvature factor and resolution of narrow regions assigned 0.5 and 0.2, respectively. Corner refinement minimum angle between boundaries were set to 240°, with element size scaling factor of 0.35. The minimum element quality value was 0.02041, average element quality was 0.3434. The number of tetrahedron was 79,491, triangles were 37,184, edge elements were 56,808, vertex elements were 580, boundary elements were 1,737,835. The overall elements were 11,168,570.

### 2.9. Numerical Methods and Governing Equations

As earlier stated, COMSOL Multiphysics has been adopted in this study on the assumption of finite element method. Species residual source, charge, momentum and mass determines convergence. The results have been tested for grid size independence. The multigrid sweep type was assigned to SSOR. The factor of error estimate was set to 20, with error ratio bound set to 0.5, and the residual factor set to 1000 [52]. The velocity and pressure damping factor was 0.5, and that of species mass fraction and current was set to 0.7 and 1, respectively. The maximum number of refinements was set to 15 in all the studied items.

Charge transfer is dependent to the species concentration, and it is well presented by Bulter-Volmer Equation (3) [52]: (3)$$j_{loc}=j_{o}\left\{exp\left(\frac{2\alpha_{a}F}{RT}\eta_{c}\right)-\left(\frac{C_{O_{2}}}{C_{O_{2},\;ref}}\right)exp\left(\frac{\left(-\alpha_{c}\right)F}{RT}\eta_{c}\right)\right\}$$
where $j_{o}$ is the exchange current density, $C_{O_{2}}$ and $C_{O_{2},ref}$ are the local and reference oxygen concentration, respectively. $\alpha_{i}$ is the transfer coefficient with respect to the different electrodes, $\eta_{c}$ is the cathode overpotential defined in Equation (4) below, *F* is the Faraday constant (96,485.332 C mol^−1^) and R is the gas constant.
(4)$$\eta =\phi_{s}-\phi_{m}-V_{oc}$$
(5)$$\nabla \cdot \left(-\sigma_{s}\nabla \phi_{s}\right)=S_{s}$$
(6)$$\nabla \cdot \left(-\sigma_{m}\nabla \phi_{m}\right)=S_{m}$$
where $\phi_{s}$ and $\phi_{m}$ stands for potentials of the electrochemically conductive solid and through the electrolyte, respectively. $V_{oc}$ stands for the electrode reference OCV. $S_{i}$ stands for the source terms.

Energy conservation for the species transport is presented by Equation (7)
(7)$$\nabla \cdot \left(\rho c_{p}uT\right)=\nabla \cdot \left(k^{eff}\nabla T\right)+S_{T}$$
where *T* is temperature, $c_{p}$ is specific heat at constant pressure, ρ is the mixture density, $k^{eff}$ is the materials effective thermal conductivity, and $S_{T}$ is the thermal source term in the respective electrodes.

Brinkman equation is used to present the species conservation, Equations (8) and (9) [32,53]:(8)$$\nabla \cdot \left(\rho u\right)=S_{m}$$
(9)$$\frac{\rho }{\varepsilon }\left(\left(u\cdot \nabla \right)\frac{u}{\varepsilon }\right)=\nabla \cdot \left[-P+\frac{\mu }{\varepsilon }\left(\nabla u+{\left(\nabla u\right)}^{T}\right)-\frac{2\mu }{3\varepsilon }\left(\nabla \cdot u\right)I\right]+S_{u}$$
where *u* is the superficial velocity, and *P* is the pressure, μ is the viscosity, ε is the porosity, *u* is the momentum term.

Maxwell-Stefan model present the mass flow through the Channels, GDL, and electrodes, Equation (10):(10)$$\nabla \cdot \left[-\rho w_{i}\overset{N}{\underset{j=1}{\sum }}D_{ij}\left\{\frac{M}{M_{j}}\left(\nabla w_{j}+w_{j}\frac{\nabla M}{M}\right)+\left(x_{j}-w_{j}\frac{\nabla P}{P}\right)\right\}+w_{i}\rho u\right]=S_{i}$$
(11)$$\rho =\frac{\left(M\right)P}{RT}$$
(12)$$M={\left(\sum \frac{w_{i}}{M_{i}}\right)}^{-1}$$

Flow rate in the anode and cathode inlets is denoted by Equation (13):(13)$$\dot{m_{n}}=\frac{I_{ref}A}{XFw_{i_{2}}}M_{i}\lambda_{n}$$
where *i* are the species, *n* denotes the cathodes and anode.

## 3. Results

### 3.1. Parameter Validation

As aforementioned, the modeling parameters have been experimentally validated. The experimental results are compared with that of the thin electrode model, with 9.825 S m^−1^ conductivity. The validation results presented in Figure 6, show negligible discrepancy which is brought by unknown influence presented by the actual conditions. The cell behavior follows the same trends over the ohmic loss region. It only varies over the kinetic and concentration loss region. 

### 3.2. Impedance and ICR

ICR is more dominant in stack cells. This is because the cells component morphology come with some defects, which render perceptible effects on the cell performance as the stack size increases. ICR depends on the contact area, Equation (14) [54]. This study is less focused on ICR.
(14)$$R=\frac{VA_{s}}{I}$$
where *R* is the electrical contact resistance, *V* is the voltage across the conducting sample, *I* portrays the applied current, and *A_s_* portrays the contact surface area. 

High ICR storm from the bipolar plate uneven morphology which is brought about by the plate surface finish and thinning during plate fabrication. These have been improved through coating. To improve the cell ICR, the compression load had to be increased during assembly. Special attention is required, as too much compression may compromise the cell porosity. 

Under normal circumstances, electrolyte resistance is proportional to its thickness, and hydration. 

In this study, we analyzed impedance spectra through a modeling approach. The impedance is simulated under four conductivity (*k*) conditions (1, 10, 100 and 1000 S/m). Although the discrepancy is minimal, the thick electrodes (thin electrolyte) cell loop is slightly wider over the imaginary axis at high conductivity (*k*), compared to the thin electrode. With decreasing *k*, the loop increases over the real axis, Figure 7a,b. 

The thin electrode cell, Figure 7a, portrays a slightly lower impedance, although it follows the same trends as that of the thick electrode cell, Figure 7b. These is brought by the long ionic transfer distance rendered by the thicker electrode case. Under the simulated electrolyte geometry, it is noted that under the 30-microns range, the most influential impedance parameter is the electrode thickness. Comparing the impedance figures (Figure 7) with that of the IV curve (Figure 8), it is clear that although the thick electrodes cell has increased impendency, Equation (15), it still shows better overall performance. This is irrefutably due to the improved reaction rate. 

Current passing through a media depends on the media’s cross-section [55,56]. Therefore, the larger the cross-sectional area, the more electrons can pass per unit time. This is portrayed by Pouillet’s law, Equation (15):(15)$$R\propto \frac{L}{A}\Rightarrow R=\rho \frac{L}{A}$$
where *R* is resistance, *L* is the conductor length, *A* is the conductor area, and ρ is the conductor resistivity.

Incorporating the influence of temperature, Equation (15) transforms to Equation (16) [57].
(16)$$\frac{dR}{dT}=\frac{d}{dT}\left(\frac{\rho L}{A}\right)=\frac{d\rho }{dT}\frac{L}{A}+\frac{dL}{dT}\frac{\rho }{A}-\frac{dA}{dT}\frac{\rho L}{A^{2}}$$*T* is temperature. The term (*dρ*/*dT*) (*L*/*A*) denote the resistance change because of the temperature dependence on the resistivity of the conductor, (*dL*/*dt*) (*ρ*/*A*) − (*dA*/*dt*) (*ρL*/*A*^2^) denote the influence of thermal expansion.

Even at low conductivity, the thick electrode cell shows high impedance to that of the thin electrode. This is attributed to the fact that the electrode thickness is more conspicuous than that of the electrolyte. This makes it impedance effect to surpass that of the electrolyte. At low frequency, the impedance discrepancy is not as pronounced as under high frequency, although it follows the same trends. 

At low k, the thin electrolyte (membrane) cell shows less gap between 0 and its initial point, compared to the thick electrolyte. 

With an increase in conductivity, the semi-circle shifts to the right over the real axis, while there is less discrepancy over the image axis. The arc size is the same in both cases. The thin electrode has a bit wider arc compared to that of the thick electrode, confirming high impedance, although the difference is not that thick electrode. At high frequency, the loops are small and expand with frequency increment. The nature of the loop suggests that, even though the discrepancy is minimal, the thin electrode cell shows less bulk electrolyte resistance and equivalent series resistance. With decreasing frequency, both cells show increasing electrolyte/electrode resistance. This might result from the electrodes porosity. The same trends confirm similar porosity in both cells. It is no clear what might have triggered the increased charge transfer in the thin electrode cell. In fuel cell, at lower frequency, the impedance does not obey the infinite Warburg equation because the flow structures usually restrict the diffusion layer to the thickness of the electrodes through convective mixing. We therefore adopt porous bounded Warburg model, Equation (17). At high frequencies, the porous bounded Warburg impedance is small since diffusing reactants do not have to move for long, although at low frequencies the reactants must diffuse further, thereby increasing the Warburg impedance. Therefore, the cell electrode should not be too thin as this may present minimal water retention and minimal catalyst content, thus affecting the cell performance. The catalyst layer thickness has significant influence in the cell impendence [58]. The membrane should remain thin.
(17)$$Z=\frac{\sigma_{i}}{\sqrt{w}}\left(1-j\right)\tan h\left(\sqrt[{\delta }]{\frac{jw}{D_{i}}}\right)$$
where j is current density, w is radial frequency, δ is diffusion layer thickness, $D_{i}$ and is the diffusion coefficient of species.

### 3.3. Polarization I-V Curve

The polarization curve portrays that the thick catalyst layer cell possesses better results than the thin catalyst layer, Figure 8. This is brought about by the improved reaction rate over the TPB, facilitated by the catalyst. Even when increasing the electrolyte conductivity of cell 1, the performance is still below that of the cell with thick catalyst layer. This study was conducted to determine the influence of electrolyte and electrodes thickness influence on the cell performance. Studies have proven that thick electrolyte increases resistance to ionic conduction. Although the thick catalyst layer comes with better results, the cost of the currently used platinum makes it unfavorable for the fuel cell industry. Cheap catalyst material can eradicate this challenge. Another challenge that comes with thick catalysts is poor generated water removal. This is because the triple phase boundary is dense and less porous; as a result, its water retention is high. The electrochemically generated water will have to travel a long distance. Therefore, it is believed that over a high current density and prolonged operation time, the generated water will end up blocking the oxygen pathway, resulting in a dramatic performance drop. Increasing the electrodes thickness, increases the dead-end pores, [59]. In this study, dead end pores are not considered. The region dominated by the ohmic losses shows the same gradient for both the thin electrode and low conductivity cell, and thick electrode cell. This reflects that the activation loss is less dependent on the electrode and electrolyte thickness. The thick electrodes cell outstands the thin electrodes cell in all the polarization curve regions. Under actual cell conditions, the thick electrodes cell is expected to experience a drastic voltage drop over the mass transfer loss region. Increasing the thin electrode conductivity to 49.825 S/m reduces the ohmic loss by 8%, although it is still lower than that of the thick electrode cell. The results in the polarization curves reveal that conductivity has less effects over the kinetic loss region, but reaction rate is effective. When the two cells are run at the same conditions, with similar conductivity, they present around 10% discrepancy. 

As seen in the impedance spectra, although the thick electrode cell renders improved reaction rate, it also renders an increased resistance. Again, even though a thin electrolyte is beneficial to the cell performance, because it comes with reduced ionic conduction resistance, it also renders some drawbacks, such as gas crossover, which increases the cell impedance [60,61], and mechanical instability. The graphs didn’t show the S shape, to portray the three losses. We believe that this resulted from the high operating temperatures (180 °C). However, Das et al. [28] got the same cell behavior with ionic increment, their cell response to conductivity was not proportional to that of this study.

### 3.4. Species Concentration

#### 3.4.1. Oxygen

H-TPEMFC mass transport is complex. It depends on numerous factors, such as viscosity, drag force, porosity, tortuosity, component hydrophobicity, and temperature. These factors definitely affect the cell performance. The level of minimum reactants in Figure 9a–c reflects the level of reaction in the cell. It is measured in the cell outlet. The inlet reactants quantity and quality are set to be the same in all the models. It is expected that the only reactants consumption occurs through electrochemical reactions as it is assumed that there is no leakage, or crossover in the cell. Therefore, the fluid leaving the cell is expected to have the lowest reactants possible; otherwise the reactants will be wasted. With purge-controlled cells, this plays a vital role in the reactant storage tank size, as it has to be as small as possible to save space and weight while being sufficient to fuel the cell for the required duration. The reactants have to be well utilized. Considering the requirements to minimize weight and size, researchers are developing open cathode cells. with open cathode cells, the cathode is fed ambient air, which contains about approximately ±21% oxygen. The reaction rate is expected to decline as you go downstream due to mass transport losses. This is because the water level increases downstream even under well drainage. The thick electrode cell outlet oxygen is 0.89 mol m^−3^ lesser than the thin electrode cell. This confirms more reaction. In the figures below, the flow direction is indicated by the arrows.

#### 3.4.2. Hydrogen

The hydrogen gas behavior (Figure 9c–e) is the sambas that of oxygen gas. The only difference lies in the axial flow direction. In this study, counter flow has been adopted as it is believed to render improved performance compared to unidirectional flow. Additionally, the quantity of inlet fuel is less than that of the reduction agent. In this study, hydrogen is only assumed to be lost through purging, with no crossover. The purge rate mainly depends on the electrolyte membrane stability. A strong membrane can accommodate lower purging rate. The thick electrode cell outlet hydrogen is 0.04 mol m^−3^. Lesser than that of thin electrode cell. 

#### 3.4.3. Electrochemically Generated Water

The generated water confirms the results reflected by the polarization curve. It is worth noting that generated water only exists in vapor form due to the high temperature. The cell with a much thicker catalyst layer shows a high level of maximum generated water (3.71 mol m^−3^) on the cathode side, Figure 10a. This confirms high level of reaction. It is worth mentioning that this may not always be the case. The high water level may also reflect poor drainage of generated water, as the water has to travel for a long route compared to thin electrodes cells. These effects are more obvious over the mass flow losses region. The water concentration increase from inlet to outlet, because the electrochemically generated water collects downstream. The cathode minimum water level is the same (0.98 mol m^−3^) in all the cases because the inlet fluid is uniform. The thick catalyst layer cell demands much attention in regard to water drainage; otherwise, the cell will be water logged. A water level discrepancy is not noted on the anode side because under normal circumstances, no water should be generated there, but only oxidation is taking place. Reduction only happens in the cathode electrode. The oxygen combines with the hydrogen ion that has been conducted through the electrolyte and the hydrogen electrons that have been conducted through the external media to the cathode side to form water. However, in other cases, it is possible for reduction to take place on the anode side, depending on the nature of the electrolyte. Under that condition, water will be generated on the anode side. It is not a surprise that the two cells with thin electrolytes in Figure 10b,c both show the same level of electrochemically generated water, which is 2.3 mol m^−3^. This is because the conductivity that is improved in those cells has nothing to do with the reaction.

## 4. Conclusions

In this study, it is revealed that the catalyst layer has a great positive influence on the cell performance. Therefore, substituting the existing platinum by developing a cost effective catalyst material can render a great improvement in the fuel cell fraternity. The thick electrode has 1.42 mol m^−3^ more electrochemically generated water compared to the thin electrode. The oxygen and hydrogen exiting the cell is 3.73 mol m^−3^, and 25.74 mol m^−3^, respectively, which is more than that of the thin electrode. This confirms better reaction rate. Even though the ionic conduction was improved by the electrolyte reduction, and conductivity improvement, the results are still below those of thick catalyst layer cell. Although the thick catalyst layer cell produces better performance in terms of reaction, there are still worries that increasing the catalyst layer will cause poor cell drainage because of the electrodes’ low porosity. In actual situations, increasing the electrode will create more blind pores. For the chosen electrolyte thickness difference (30 μm), the resulting overall impedance proves to be less dependent of the electrolyte thickness, but significantly dependent on the electrodes thickness. At low conductivity, the impendency discrepancy is more discernible in relation to the electrodes thickness. For all the cases, impedance under high conductivity seems the same. The polarization curve shows that the thick electrode cell has 10% better performance than the thin electrode cell. The aforementioned result indicate that H-TPEMFC can accommodate much thicker electrode as most of the fluid involved is in gas form, therefore it can penetrate through electrode at a fast rate compared to liquid fluid.

## Figures and Tables

**Figure 1 membranes-13-00012-f001:**
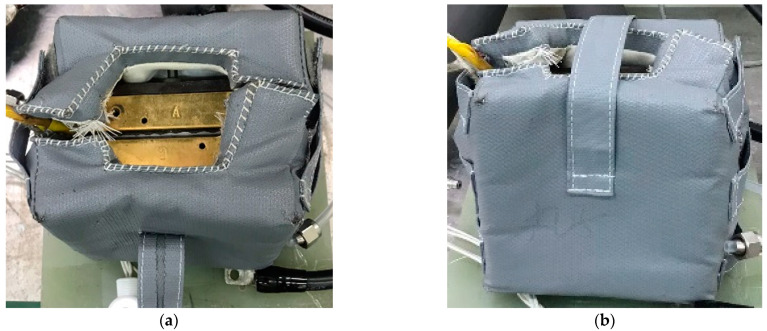
Covered cell to prevent heat loss, (**a**) top view, and (**b**) side view.

**Figure 2 membranes-13-00012-f002:**
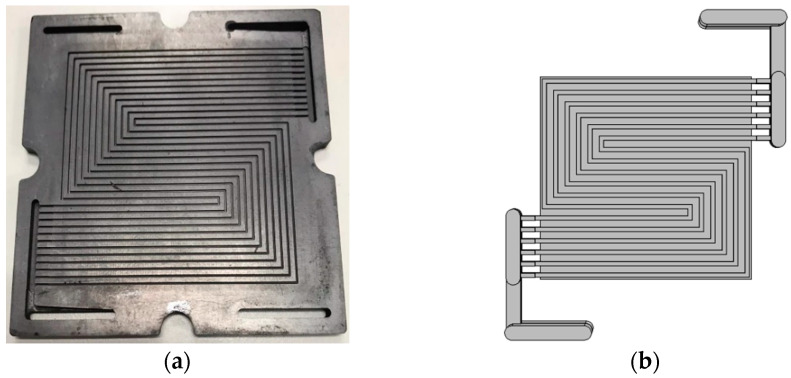
Serpentine channel schematic, (**a**) model experimental channels, and (**b**) modeling geometry.

**Figure 3 membranes-13-00012-f003:**
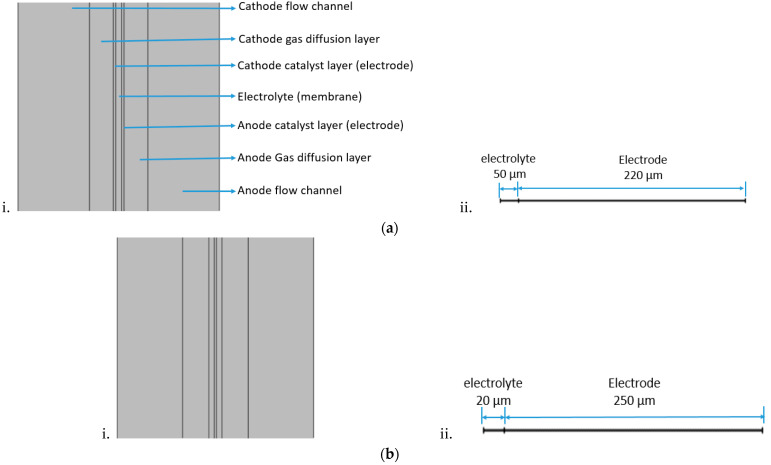
Schematic diagram of (**a**) i. thin electrode cell, ii. 1D half-cell used in impedance simulation (**b**) i. thick electrode cell, ii. 1D half-cell used in impedance simulation.

**Figure 4 membranes-13-00012-f004:**
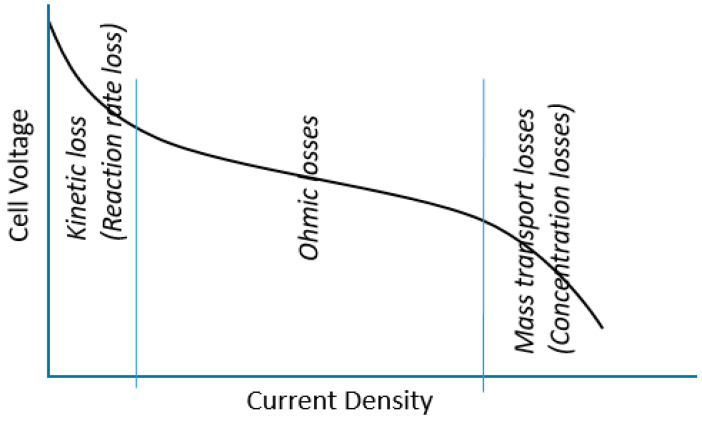
Schematic of PEMFC voltage losses.

**Figure 5 membranes-13-00012-f005:**
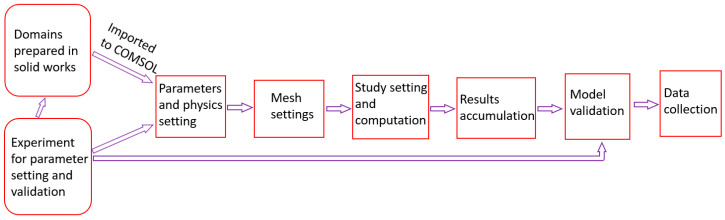
Study flowchart.

**Figure 6 membranes-13-00012-f006:**
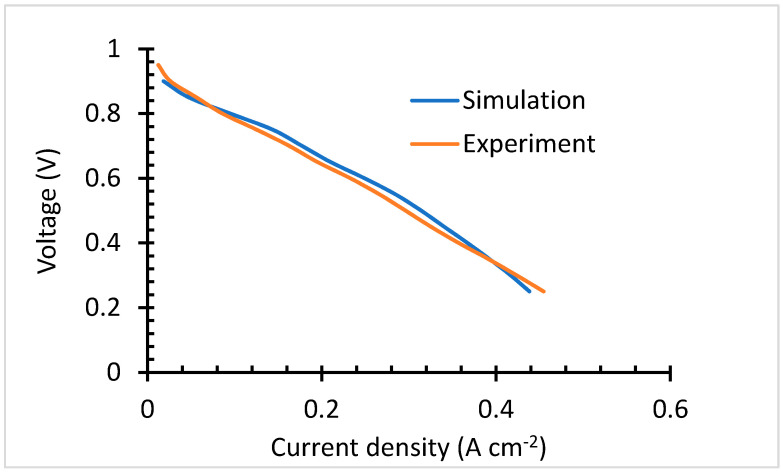
Parameter validation.

**Figure 7 membranes-13-00012-f007:**
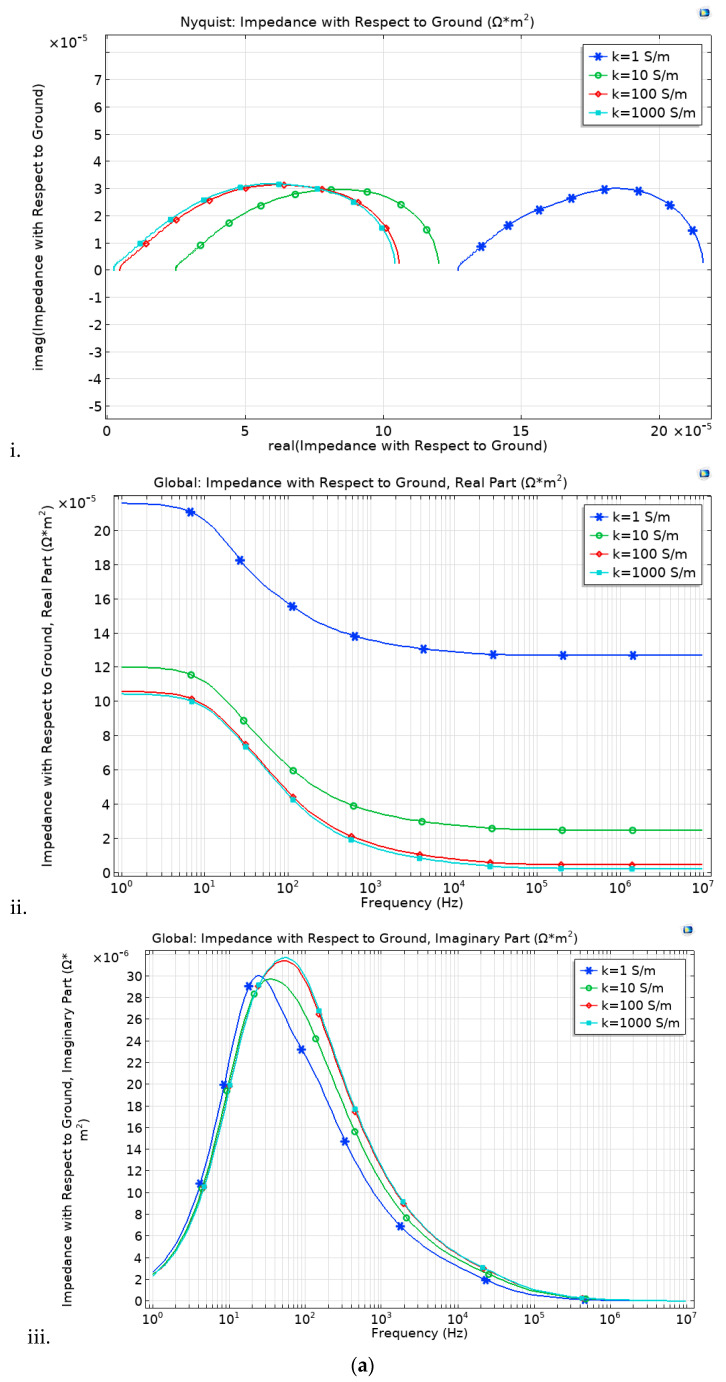
Impedance spectroscopy with respect to the electrode and electrolyte thickness, (**a**) thick electrode (250 μm) and (**b**) thin electrode (220 μm).

**Figure 8 membranes-13-00012-f008:**
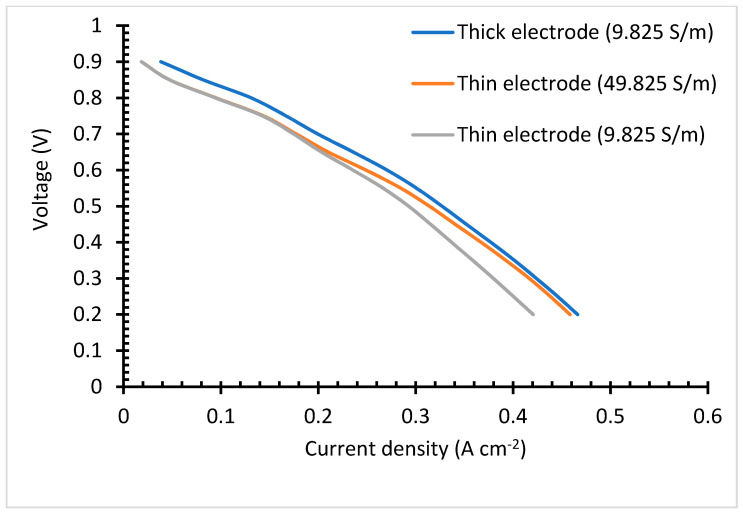
Polarization curve.

**Figure 9 membranes-13-00012-f009:**
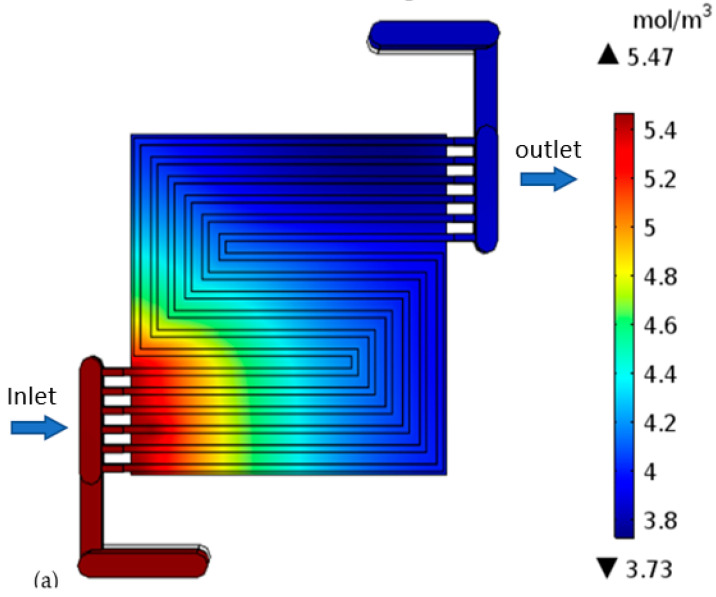
Oxygen concentration in the cell with: (**a**) thick electrode, (**b**) thin electrode (9.825 S/m) and (**c**) thin electrode (49.825 S/m); Hydrogen concentration in the cell with: (**d**) thick electrode, (**e**) thin electrode (9.825 S/m) and (**f**) thin electrode (49.825 S/m).

**Figure 10 membranes-13-00012-f010:**
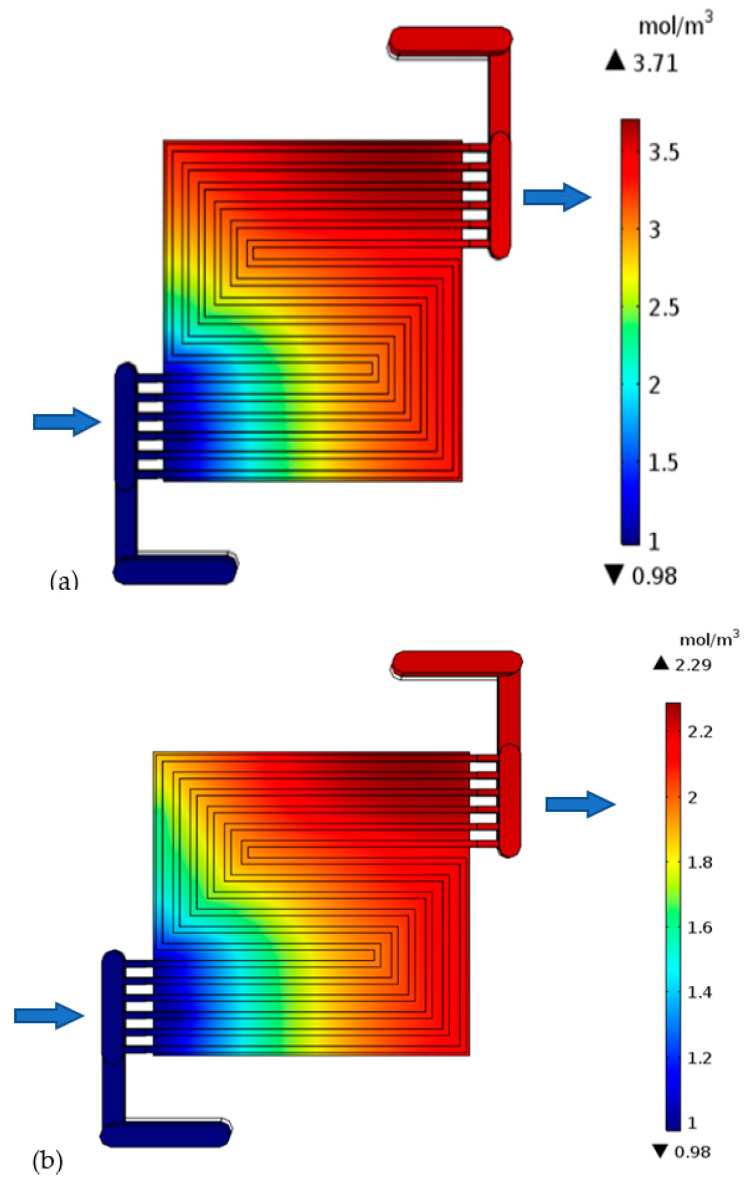
Generated water in the cathode side of the cell with: (**a**) thick electrode, (**b**) thin electrode (9.825 S/m) and (**c**) thin electrode (49.825 S/m).

**Table 1 membranes-13-00012-t001:** Experiment and simulating cell geometry.

Items	Cell 1 (Thin Electrode)	Cell 2 (Thick Electrode)	Cell 3 (Cell 1 with Improved Conductivity)
Electrolyte thickness (μm)	50	20	50
Electrode (μm)	20	50	20
GDL (μm)	200	200	200
Electrolyte Conductivity (S/m)	9.825	9.825	49.825
Electrode Conductivity (S/m)	222	222	222
Catalyst Porosity (%)	35	35	35
GDL Porosity (%)	60	60	60
Channel width (mm)	1	1	1
Channel depth (mm)	1	1	1
Rib width (mm)	0.6	0.6	0.6

**Table 2 membranes-13-00012-t002:** Modeling parameters.

Item	Values
Operating temperatures (°C)	180
Anode stoichiometry	1.2
Cathode stoichiometry	3
Reference pressure (Pa)	101,325
Anode velocity (m/s)	0.2
Cathode velocity (m/s)	0.5

## Data Availability

Not applicable.

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
