# Peer review of "Analyses of the Effects of Electrolyte and Electrode Thickness on High Temperature Proton Exchange Membrane Fuel Cell (H-TPEMFC) Quality"

_membranes, 2022, doi:10.3390/membranes13010012_

Round 1
Reviewer 1 Report
I think that the results in this paper will be useful to design practical electrodes of PEMFCs. However, I also think that the authors should improve the quality of the manuscript. Please discuss and modify the following points.
1 (Page 3, Line 122)
What does “No specie cross-over” mean? Is this the mistake of “No species cross-over”?
2 (Page 7, Line 220)
What does the thermal stability (200 oC) written in your paper mean? As far as I know, the glass temperature of Nafion is about 100oC.
3 (Page 7, Line 236, 237)
The proton conductivity σ and gas constant R should be written in italic characters.
4 (Page 9, Line 284-285)
I think that the sentence (“As aforementioned, the modeling parameters have been validated, and the validation results are presented in Fig. 5 show negligible discrepancy.”) is wrong in the viewpoint from grammar. Did you want to write the sentence “As aforementioned, the modeling parameters have been validated, and the valida-284 tion results are presented in Fig. 5, showing negligible discrepancy.”?
5 (Figure 5)
You should denote the superscript to denote the unit of current density (A/cm2).
6 (Page 9, Line 284-285)
Why are Experiment and Simulation in Figure 5 described as the sloping straight lines of instead of S-curves shown in Figure 4? Can you evaluate three losses (kinetic, ohmic, and mass transport) correctly?
7 (Page 10, Line 355-356)
What could you suggest from the small and expanded loops at high frequency?
8 (Figure 7)
Please modify the three points: (1) The label of “overpotential” is overlapped with the number of the vertical axis. (2) You should denote the superscript to denote the unit of current density (A/cm2). (3) What the grey line represent? Please add the label in Figure 7.
9 (Page 16, Line 432-439)
Could you show the hydrogen concentration in the figures similar to Figure 8? I want to understand the detail.
10 (Reference)
Where did you use reference 48 and 49?
Author Response
Dear Dr.
Thank you so much for the constructive comments. These comments will not only be used in this manuscripts, but they will further help us improve our future work. We have tried our best to respond to the comments. A blue color is used to address the comments, unless stated. In the manuscript, red color is used for major changes.
Please find the response to the comments and the manuscript draft attached.
Kind regards.
Mangaliso Menzi Dlamini

Reviewer 2 Report
Manuscript Number: membranes-2037658
Title: Analyses of the Effects of Electrolyte and Electrode Thickness on High Temperature Proton Exchange Membrane Fuel Cell (H-TPEMFC) Quality
This paper reported on the study effects of electrolyte and electrode thickness on high temperature proton exchange membrane fuel cell (H-TPEMFC) performance. In general, the current content of this manuscript had some drawbacks especially on the novelty of the study and detail discussion in the introduction, methods and finding parts. Therefore, an improvement is needed for this current manuscript and authors should be paying special attention to the following comments;
1. English need to be improved. Please also check the spelling.
2. Abstract: Abstract should be written in concise and precisely. Both, quantitative and qualitative finding should be highlighted.
3. Introduction: Lack of critical discussion on the research gap for the mathematical model and computational fluid dynamic (CFD) for H-TPEMFC. Therefore, a comprehensive discussion are need based on previous research works by other researchers should be highlighted.
4. Introduction: The novelty of this study is not clear. Therefore, it is suggested to provide more discussion/ background of this study based on previous research works to emphasize the novelty of this study.
5. Section 1.1: Is that referring to the experimental work for this study? If yes, it is suggested to reallocate in the material and methods section. In addition, the experimental work should be clarified in detail.
6. Material and methods: The modelling/ simulation should be written in concise and precisely (need to be simplified). It is suggested to provide a flowchart for this modelling/ simulation. Several important equations should be provided in the main text.
7. Model development: What are the assumptions for the model development?
8. Figure 6: Why only two k values have been investigated? How about the profile for k = 1 and k=100. Please clarify in detail.
9. Figure 5 and 7: The figure need to be improved. For instance cm^2 should be changed to cm2 and overpotantial vs overpotential.
10. Figure 7: What is big and small electrode? The legend is not specifically representing the intended meaning of discussion. Is that referring the thickness of electrode?
11. 3.4. Species concentration: Is there any comparison with the previous finding by other researchers?
12. Figure 8 and Figure 9: It is suggested to provide the label for the input and output of the fuel/ reactant.
13. Conclusion: Conclusion should be written in concise and precisely. Both, quantitative and qualitative finding should be highlighted.
Author Response

(The authors gave the same response as above.)

Reviewer 3 Report
The authors investigated the effects of the electrode and electrolyte thickness on the high-temperature proton exchange membrane fuel cell performance. However, the method and findings of the manuscript were not innovative enough compared to previous publications.
My detailed comments as follow:
1, I don’t not clearly understand the contribution of the paper. Many ideas were discussed in previous researchers. It would be nice that authors can build and apply a framework to solve a set of specific application problems. Unfortunately, I didn’t get that in this version
2,The original creativity and academic significance of this manuscript are not clear, and the contribution of this work should be further clarified in Introduction part.
3,It is noted that your manuscript needs careful editing by someone with expertise in technical English editing paying particular attention to English grammar, spelling, and sentence structure so that the goals and results of the study can be more clearly delivered to the readers.
Author Response

(The authors gave the same response as above.)

Round 2
Reviewer 2 Report
In overall, the authors had answered and clarified all the comments/ suggestions in a good manner and were stated clearly in the revised text version. Taking this aspects into account I suggest that the manuscript should be accepted for publication in the membranes.
Reviewer 3 Report
The paper has been revised prudently and throughly, which can be publieshed in Membranes.